# Matching Supply and Demand with Lead-Time Dependent Price and with Safety Stocks in a Make-to-Order Production System

Sonu Kumar Das * and Thyagaraj S. Kuthambalayan

Department of Management Studies, Indian Institute of Technology (ISM), Dhanbad 826004, Jharkhand, India
* Correspondence: skdas371.16dr000218@ms.iitism.ac.in; Tel.: +91-9771129693

**Abstract:** We studied the ability to reduce the supply–demand mismatch of a periodic Make-to-Order (MTO) production system using safety stocks with marketing managing demand using lead-time guarantee and price as levers. The aim is to understand the interdependencies between lead-time guarantee, price, and safety stocks. We modeled the problem as an unconstrained stochastic non-linear programming problem, maximizing the expected profit per-unit time and obtaining a closed-form solution. The price is a function of the lead-time guarantee. Based on the sensitivity analysis of problem parameters, we found that lead-time competitiveness is adversely affected by a low safety stock level, MTO production rate (i.e., low supply capability), and product price (i.e., high demand volume). A shorter lead-time requires higher safety stock through reduced product and inventory holding costs. A higher price for a shorter lead-time in a lead-time-sensitive market reduces the safety stock. In a price-sensitive market, lead-time is decreased instead of the price. Demand variation results in longer lead-time and higher safety stock (provided the holding cost is low). For a higher price premium, price increases and lead-time decrease (safety stock increases). The integrated operation-marketing model captures the complex trade-offs not seen in a hierarchical model to produce better solutions.

**Keywords:** operations-marketing model; guaranteed lead-time; price; make-to-order; make-to-stock

## 1. Introduction

The supply–demand mismatch is the result of inadequate supply and demand co-ordination. It is due to poor supply performance (quality, delivery) of the (1) upstream supplier(s), (2) local firms, and (3) downstream distributors/retailers and is exacerbated by the uncertainty of customer(s) demand [1]. One of the basic approaches to reducing supply–demand mismatch is product management through postponement (e.g., make-to-order (MTO), assemble-to-order (ATO), and process sequencing). The MTO production system produces only in response to confirmed orders and thereby finds it challenging to compete on delivery lead-time with an increase in load. A 'periodical' MTO production system is used if the aggregate demand is continuous [2].

Demand, information, and supply management are other primary approaches to reducing supply–demand mismatch [3,4]. Firms often advertise or guarantee a lead-time within which they expect to meet the demand with high probability and make every effort to ensure it is competitively short yet reliable. Reliable lead-times reduce the need for change in production plans and are preferred by customers over a short but unreliable lead-time. Lead-time guarantees serve to manage demand to obtain an improved match with the firms' supply.

The reliability of the lead-time guarantee by the marketing needs to be ensured by a suitable operations strategy. The integrated firm that coordinates well between departments is in the best position to avoid conflicts and make good the lead-time promise [5,6].

An integrated operations-marketing approach with effective 'information management' is the key to reducing supply–demand mismatch.

In response to demand uncertainty, firms produce the less-uncertain portion of product demand to stock (which we term safety stock) using an efficient, low-cost specialized manufacturing facility (referred to in this study as a secondary supply source). The more uncertain portion of product demand is produced using a high-cost flexible manufacturing facility based on firm order. The 'efficient' plant focuses on cost reduction by improving resource usage. The 'flexible' manufacturing system competes on lead-time at the expense of cost. This strategy is also termed the 'focus' strategy via 'space' separation [7]. Supply management using safety stock in an MTO production system improves operational flexibility to reduce supply–demand mismatch.

### 1.1. Research Motivation

Several individual industrial situations, which have been documented in operations management literature, have motivated this research. This work was partly motivated by a leather furniture manufacturer's industrial situation, described in Rao et al. [8], which guarantees retailers delivery within two weeks. Offering a lead-time guarantee, which is uniform to all customers, is also usual for many firms engaged in construction, industrial equipment supply, catering, and transportation. Lead-times determine the success of such systems with little or no finished goods inventory [9]. A shorter lead-time attracts more demand, as seen in the situation of a food packaging film manufacturer described by Dobson and Yano [10]. Besides a shorter lead-time, a smaller price quote attracts more demand [11].

A large demand volume makes it more likely that the firm shall fail to meet the quoted delivery time. Safety stocks in an MTO environment cater to the element of demand, which can be well-forecasted, and MTO strategy to the element with high forecast error. Using safety stock in an MTO setting has long been employed in the fashion industry [12,13]. It serves to improve lead-time competitiveness. Moreover, industry practice, especially prevalent among Web retailers (e.g., Amazon, Flipkart), suggests that customers are willing to pay a price premium for a shorter lead-time [6,14,15].

### 1.2. Objective and Approach

We studied the ability to reduce supply–demand mismatch of a periodic MTO production system using safety stocks with marketing managing demand using the two product attributes (i) lead-time guarantee, and (ii) price as levers. We modeled the problem as an unconstrained stochastic non-linear programming (NLP) problem of two-stage, maximizing the expected profit per-unit time. Stage 1 decisions are taken before demand realization. In Stage 1, operations decide the level of safety stocks and length of the production planning cycle, thereby managing supply. Marketing decides the two product attributes, thereby managing demand. In Stage 2, demand is realized and is modeled as a linear function of the product attributes, with a non-negative stochastic error component. The price is explicitly modeled as a function of the lead-time guarantee. We provide a closed-form solution using the multivariable optimization technique [16,17]. The sensitivity of demand and cost parameters and production rate on decision variables and expected profit per-unit time is performed, and essential managerial insights are drawn.

### 1.3. Research Contribution

We explicitly specify the dependencies between price and lead-time in an integrated operations-marketing model to capture the trade-offs between lead-time guarantee, price, and safety stock. We find that lead-time competitiveness is adversely affected by a low safety stock level, MTO production rate (i.e., low supply capability), and product price (i.e., high demand volume).

1.　To promise a short lead-time, an integrated firm aims to increase safety stocks by reducing product costs and setting a low inventory holding cost.

2. A higher price quote corresponding to a shorter lead-time in a lead-time-sensitive market reduces the need to increase the safety stock level. In a price-sensitive market, the firm would reduce lead-time rather than the price.

3. Demand variability is countered by guaranteeing a longer lead-time and increasing the safety stock (The stock level is lowered when the holding costs are high).

4. The firm sets a higher product price and quotes a shorter lead-time (thereby increasing the safety stock level) in response to an opportunity to charge a higher price premium for a given lead-time decrease.

The integrated operation-marketing model captures the complex trade-offs not seen in a hierarchical model to produce better solutions.

### 1.4. Literature Review

This study considers decisions on two product attributes: (i) lead-time guarantee and (ii) price, and decisions on production capacity (production cycle) and safety stock level in a periodic MTO production system. Few studies consider the effect of decisions on both price and lead-time guarantee, on-demand volume. Dobson and Yano [10] modeled demand as lead-time and price dependent in a mixed MTS-MTO system to evaluate a product-wise MTS vs. MTO decision. Ray and Jewkes [14] modeled customer's product switchover decision owing to price and a lead-time difference between two products. Rao et al. [8] modeled demand as a function of lead-time guarantee in an MTO system and evaluated how production cycle length relates to the lead-time guarantee. Pekgün et al. [18] designed a contract to coordinate decentralized price (marketing) and lead-time (operations) decisions. Shao and Dong [19] modeled the random demand as a linear function of lead-time in an ATO system and evaluated back-ordering, backup sourcing, and compensation strategy as a response to supply disruption. Qian [20] modeled product demand as a deterministic linear function of service level and quality besides price and guaranteed lead-time to evaluate suppliers in the MTO system. Kuthambalayan et al. [21] modeled demand as a linear function of guaranteed lead-time and price to capture the trade-offs of inventory of components, semi-finished goods, and outsourcing costs in the ATO system. Pekgün et al. [22] studied cross-price and lead-time effects with decentralized capacity, price, and lead-time decisions. Kuthambalayan and Bera [23] modeled demand as a function of guaranteed lead-time in a mixed MTO/MTS production system with level dependency to evaluate a product-wise MTS vs. MTO decision.

Additionally, few studies consider both safety stock and MTO strategy to manage the supply. Eynan and Rosenblatt [24] studied an ATO system and Assemble-in-Advance (AIA) system and evaluated the benefits of the production system in terms of reduced overstocking risk compared to a pure AIA system and improved lead-time service in comparison to an ATO system. Eynan and Rosenblatt [25] confirmed the benefits in an assembly system with component commonality. In contrast, Hariga [26] confirmed it in a multi-echelon assembly system, Fu et al. [27] confirmed it in situations where components have long procurement lead-times and the firm's production capacity is outsourced, and Xiao et al. [28] confirmed it in situations of uncertain assembly capacity. Altendorfer and Minner [29] evaluated the benefits of an MTO system with safety stocks by analyzing demand and cost parameters, production rate, and mean customer required lead-time.

Research Positioning

We used the work of Xiao et al. [28] as the starting point of our research. In their model, the demand is free from market characteristics. In contrast, we consider the demand as a linear function of two product attributes: (i) guaranteed lead-time, and (ii) price [20] with a non-negative stochastic error component. This study differs further from Xiao et al. [28] as we assume the safety stock to be supplied by a secondary source instead of being self-produced. This study also differs from Qian [20] due to the stochastic demand assumption (while they consider the actual delivery time stochastic) and the safety stock aspect. Most studies consider real-time lead-time quotation while this study determines

tactical level lead-time guarantee before demand realization. Additionally, most studies consider guaranteed lead-time mainly in an MTO system, a few in a mixed MTS-MTO production setting, but to the best of our knowledge, none in an MTO system with safety stocks. Additionally, the product price in this study is lead-time dependent as shown in the Table 1.

**Table 1.** Literature on managing demand with lead-time.

| Study | Model | Strategy | Demand | | Price | |
| | | Mixed or Safety Stock | Price | Lead-Time | Stochastic | Lead-Time Dependent |
|---|---|---|---|---|---|---|
| Dobson and Yano (2002) | NLP | Mixed | ✓ | ✓ | | |
| Ray and Jewkes (2004) | NLP | | ✓ | ✓ | | |
| Rao et al. (2005) | NLP | | ✓ | ✓ | ✓ | |
| Pekgün et al. (2008) | NLP | | ✓ | ✓ | | |
| Shao and Dong (2010) | NLP | | ✓ | ✓ | ✓ | |
| Qian (2014) | NLP | | ✓ | ✓ | | |
| Kuthambalayan et al. (2014) | MINLP | | ✓ | ✓ | ✓ | |
| Pekgün et al. (2017) | NLP | | ✓ | ✓ | ✓ | |
| Kuthambalayan and Bera (2020) | NLP | Mixed | ✓ | ✓ | ✓ | |
| This Study | NLP | Safety stock | ✓ | ✓ | ✓ | ✓ |

Further, in the paper, Section 2 provides the Materials and Methods. Section 3 provides results of a numerical example and sensitivity analysis of key problem parameters. Section 4 discusses the results of sensitivity analysis. Section 5 concludes with future scope of the research.

## 2. Materials and Methods

### 2.1. Assumptions and Notations

The firm sells a single product to customers at multiple locations in this study. The orders for products are continuously received and accumulated by the sales department over the production planning cycle $T$ before communicating to production. The marketing department pre-announces an upper-bound on lead-time $2T$. The production schedule is updated once every period $T$, and customers can adjust their orders until the end of the preceding $T$ period. At this time, their order acceptance decision is finalized, and operations make an improved allocation of orders and capacity planning. This problem set is motivated by Rao et al. [8].

We model the problem as a two-stage stochastic problem. Stage 1 decisions are taken before actual demand realization. In Stage 1, operations decide the level of safety stocks $q_2$ and the length of the production planning cycle $T$ (thereby, managing supply). The MTO production quantity is limited to $q_1$ due to the planning cycle time constraint $T$ and the production rate $\rho_1$. Additionally, in Stage 1, marketing decides the two product attributes, price $p$ and lead-time guarantee $2T$ (thereby, managing demand). In Stage 2, demand is realized and is modeled as a linear function of the product attributes, with a non-negative stochastic error component $D$, $D' = a - 2b_1 T - b_2 p + D$. Here, $a$ is the basic market size, $b_1$ is the non-negative lead-time sensitivity of demand, and $b_2$ is the non-negative price sensitivity of demand. $D$ is assumed to be nonnegative with a uniform distribution, $D \sim U(0, c)$, to capture uncertainty. $f(D)$ denotes the density function and $\int f(D)d(D)$ is the distribution function of $f(D)$ and continuously differentiable. The price $p$ is explicitly modeled as a function of the lead-time guarantee $2T$, $p = p_1 - e(2T)$. Here, $p_1$ is the selling price of the product when the lead-time guarantee $2T$ is zero and $e$ is the non-negative lead-time sensitivity of price. We summarize these and additional notation in Table 2.

In the second stage, at the beginning of the period $T$, product demand information is received. There are three possible scenarios:

1.  *The product demand is less than the safety stock level $q_2$*: The demand is met from available safety stock $q_2$ at a per-unit cost $t_2$ and per-unit selling price $p$. Any safety stock in excess of the demand quantity $D'$ incurs a per-unit time holding cost $r_3 t_2$ ($0 < r_3 < 1$). In this scenario, MTO quantity and demand shortages are nil.

2. *The product demand exceeds the safety stock level $q_2$ but is less than the planning cycle MTO limit $q_2$:* The demand is first met from available safety stock $q_2$ at a per-unit cost $t_2$ and per-unit selling price $p$. Any excess demand is made-to-order at a per-unit cost $t_1$, with $(t_2 < t_1 < p)$ [25,30], and per-unit selling price $p$. In this scenario, excess safety stock and demand shortages are nil.

3. *The product demand exceeds the sum of safety stock level and planning cycle MTO limit $(q_2 + q_1)$:* The demand is first met from available safety stock $q_2$ at a per-unit cost $t_2$ and per-unit selling price $p$. The quantity $q_1$ is then made-to-order at a per-unit cost $t_1$, and per-unit selling price $p$. In this scenario, a shortage cost of $r_2 t_1$ $(0 < r_2 < 1)$ is incurred on any demand quantity $D'$ which exceeds $(q_2 + q_1)$.

**Table 2.** Notation.

| Parameters | |
|---|---|
| $a$ | Basic market size in a linear attribute-dependent demand function (in units) |
| $D \sim U(0,c)$ | Non-negative error (in units) in estimating the market demand (stochastic component of the market demand with uniform distribution) |
| $p_1$ | Per-unit selling price (in \$) of the product when lead-time is zero |
| $r_2$ | Per$-$unit cos t of shortage (in \$) is given by $r_2 t_1$, $(0 < r_2 < 1)$ |
| $r_3$ | Per$-$unit per unit time cos t of holding is given by $r_3 t_2$ (in \$), $0 < r_3 < 1$ |
| $t_1$ | Per-unit cost of MTO product (in \$) |
| $t_2$ | Per$-$unit cos t of safety stock $(t_2 < t_1)$ (in \$) |
| $\rho_1$ | Production rate of MTO product (in units per day) |
| $b_1$ | Non-negative lead-time sensitivity of the demand (in units per day) |
| $b_2$ | Non-negative price sensitivity of the demand (in units per \$) |
| $e$ | Non-negative lead-time sensitivity of price (in \$ per day) |
| Decision variables and the objective | |
| $q_1$ | MTO production limit (in units) in time $T$ at production rate $\rho_1$ |
| $q_2$ | Level of safety stock (in units) |
| $2T$ | Uniform guaranteed lead-time (in days) |
| $p$ | Per-unit price the product (in \$) |
| $E[z]$ | Expected profit per-unit time (in \$ per day) |

*2.2. Model Formulation*

The demand function is modeled as $D' = a - 2b_1 T - b_2 p + D$. Demand increases with a decrease in lead-time and price. If $T$ increases, the delivery time will be longer. The time-sensitive customer segment may cancel or may not place the order if the delivery time is longer. Consequently, the firm's demand will decrease, which will lead to a decrease in the firm's profit. This situation can be managed by making the price lead-time sensitive $(p = p_1 - e(2T))$. This implies that the customer shall pay a lower price for a longer lead-time. To trace such a situation, since price is not an independent decision variable, demand function is expressed as $D' = a' - b'T + D$, where, $a' = a - b_2 p_1$ and $b' = 2b_1 - 2b_2 e$. The model under this condition captures the trade-off between lead-time, price, and safety stock.

The objective is to maximize expected profit per-unit of time. The expected profit over period time $T$ comprises the following components in three different scenarios over period time $T$, which are explained below:

1. The product demand is less than the safety stock $(0 \le D' \le q_2)$ This implies that $b'T - a' \le D \le q_2 + b'T - a'$

   (a) The sales revenue is $\int_{b'\frac{q_1}{\rho_1} - a'}^{q_2 + b'\frac{q_1}{\rho_1} - a'} pD' f(D) d(D)$,

(b)    The cost of products is $\int_{b'\frac{q_1}{\rho_1}-a'}^{q_2+b'\frac{q_1}{\rho_1}-a'} q_2 t_2 f(D)d(D)$, and

(c)    The holding cost is $r_3 t_2 \int_{b'\frac{q_1}{\rho_1}-a'}^{q_2+b'\frac{q_1}{\rho_1}-a'} T(q_2-D')f(D)d(D)$.

2.   The product demand exceeds the safety stock ($D' > q_2$) but is less than the planning cycle MTO limit $q_1(D' \le q_1)$: This implies that $q_2+b'T-a' \le D \le q_2+q_1+b'T-a'$.

(a)    The sales revenue is $\int_{q_2+b'\frac{q_1}{\rho_1}-a'}^{q_2+q_1+b'\frac{q_1}{\rho_1}-a'} (q_2 p+(D'-q_2)p)f(D)d(D)$ and

(b)    The cost of products is $\int_{q_2+b'\frac{q_1}{\rho_1}-a'}^{q_2+q_1+b'\frac{q_1}{\rho_1}-a'} (q_2 t_2+t_1(D'-q_2))f(D)d(D)$.

3.   The product demand exceeds the quantity $(q_2+q_1)$ $(D' \ge q_2+q_1)$: Since, $D \sim U(0,c)$, this implies that $c \ge D \ge q_2+q_1+b'T-a'$.

(a)    The sales revenue is $\int_{q_2+q_1+b'\frac{q_1}{\rho_1}-a'}^{c} p(q_2+q_1)f(D)d(D)$,

(b)    The cost of products is $\int_{q_2+q_1+b'\frac{q_1}{\rho_1}-a'}^{c} (q_2 t_2+t_1 q_1)f(D)d(D)$, and

(c)    The shortage cost is $r_2 t_1 \int_{q_2+q_1+b'\frac{q_1}{\rho_1}-a'}^{c} (D'-(q_2+q_1))f(D)d(D)$.

Expected profit over period time $T$ is the difference between sales revenue and net cost (production cost, holding cost, and shortage cost). The objective function, which is to maximize the expected profit per-unit of time is given below with $T$ replaced by $\frac{q_1}{\rho_1}$.

Substituting $f(D)=1/c$, $p=p_1-e(2q_1/\rho_1)$, $a'=a-b_2 p_1$, and $b'=2b_1-2b_2 e$, the expected profit per-unit time is given as:

$$
\begin{aligned}
E[z] = &\int_{b'\frac{q_1}{\rho_1}-a'}^{q_2+b'\frac{q_1}{\rho_1}-a'} p\frac{\left(a'-b'\frac{q_1}{\rho_1}+D\right)}{\frac{q_1}{\rho_1}}f(D)d(D) - \int_{b'\frac{q_1}{\rho_1}-a'}^{q_2+b'\frac{q_1}{\rho_1}-a'} \frac{q_2 t_2}{\frac{q_1}{\rho_1}}f(D)d(D) \\
&-r_3 t_2 \int_{b'\frac{q_1}{\rho_1}-a'}^{q_2+b'\frac{q_1}{\rho_1}-a'} \left(q_2-a'+b'\frac{q_1}{\rho_1}-D\right)f(D)d(D) + \int_{q_2+b'\frac{q_1}{\rho_1}-a'}^{q_2+q_1+b'\frac{q_1}{\rho_1}-a'} \frac{\left(q_2 p+\left(a'-b'\frac{q_1}{\rho_1}+D-q_2\right)p\right)}{\frac{q_1}{\rho_1}}f(D)d(D) \\
&-\int_{q_2+b'\frac{q_1}{\rho_1}-a'}^{q_2+q_1+b'\frac{q_1}{\rho_1}-a'} \frac{\left(q_2 t_2+t_1\left(a'-b'\frac{q_1}{\rho_1}+D-q_2\right)\right)}{\frac{q_1}{\rho_1}}f(D)d(D) + \int_{q_2+q_1+b'\frac{q_1}{\rho_1}-a'}^{c} \frac{p(q_2+q_1)}{\frac{q_1}{\rho_1}}f(D)d(D) \\
&-\int_{q_2+q_1+b'\frac{q_1}{\rho_1}-a'}^{c} \frac{(q_2 t_2+t_1 q_1)}{\frac{q_1}{\rho_1}}f(D)d(D) - r_2 t_1 \int_{q_2+q_1+b'\frac{q_1}{\rho_1}-a'}^{c} \frac{\left(a'-b'\frac{q_1}{\rho_1}+D-(q_2+q_1)\right)}{\frac{q_1}{\rho_1}}f(D)d(D) \\
E[z] = &-\frac{1}{2cq_1\rho_1}(4b_1^2 q_1^2 r_2 t_1+b_2^2 r_2 t_1(2eq_1-p_1\rho_1)^2 - 4b_1 q_1(2eq_1(q_1+q_2+b_2 r_2 t_1) \\
&-((q_1(-1+r_2)-(c+a-q_2)r_2)t_1+p_1(q_1+q_2+b_2 r_2 t_1)-q_2 t_2)\rho_1) \\
&+2b_2(2eq_1-p_1\rho_1)(2eq_1(q_1+q_2)+(-p_1(q_1+q_2)+(q_1+(c+a-q_1-q_2)r_2)t_1+q_2 t_2)\rho_1) \\
&+\rho_1(q_1(-2e(q_1+q_2)(-2c-2a+q_1+q_2)+q_2^2 r_3 t_2)+-p_1(2c+2a-q_1-q_2)(q_1+q_2) \\
&+\left(-q_1(-2c-2a+q_1+2q_2)+(c+a-q_1-q_2)^2 r_2\right)t_1+2(c+a)q_2 t_2)\rho_1))
\end{aligned}
\tag{1}
$$

**Lemma 1.** *The expected profit function $E[z]$ is concave in $q_1$ and $q_2$ under certain conditions. The proof of the lemma1 is given in the Appendix* A.

The following Proposition addresses the optimal solution of the problem through the necessary conditions of Optimization Theory.

**Proposition 1.** *In both price, as well as delivery time sensitive market, the manufacturer's optimal planning cycle MTO limit and safety stock level $(q_1^*, q_2^*)$ satisfy first order necessary conditions $\frac{\partial z}{\partial q_1}=0$ and $\frac{\partial z}{\partial q_2}=0$* [31]. *Which is given as:*

$$
\begin{aligned}
&J_1 q_1^3+K_1 q_1^2 q_2+L_1 q_2^2+M_1 q_2+N_1=0 \\
&\text{where, } J_1=(16e(b_1-eb_2)+4e\rho_1), K_1=(8e(b_1-eb_2), L_1=(p_1\rho_1^2+r_2 t_1\rho_1^2), \\
&M_1=\left(2(c+a)t_2\rho_1^2-2p_1(c+a+b_2(-r_2 t_1+t_2))\rho_1^2+2b_2 p_1^2\rho_1^2-2cr_2 t_1\rho_1^2-2ar_2 t_1\rho_1^2\right) \\
&N_1=(c+a)^2 r_2 t_1\rho_1^2+b_2 p_1^2 b_2 r_2 t_1\rho_1^2-2(c+a)b_2 p_1 r_2 t_1\rho_1^2+(b_1-eb_2)(-b_1+eb_2)4r_2 t_1-4ce\rho_1 \\
&-4ea\rho_1+4eq_2\rho-4b_1(p_1+(-1+r_2)t_1)\rho_1+2eb_2(2p_1+(-1+r_2)t_1)\rho_1-p_1\rho_1^2+(1-r_2)t_1\rho_1^2)
\end{aligned}
\tag{2}
$$

$$J_2 q_1^2 + K_2 q_1 q_2 + L_2 q_2 + M_2 q_1 + N_2 = 0$$
$$\text{where, } J_2 = (4e(b_1 - eb_2) + 2e\rho_1), K_2 = (2e\rho_1 - r_3 t_2 \rho_1), L_2 = -(r_2 t_1 \rho_1^2 + p_1 \rho_1^2), M_2 = -2ce\rho_1$$
$$-2ea\rho_1 + 2eb_2(2p_1 + r_2 t_1)\rho_1 - 2b_1(p_1 + r_2 t_1 - t_2)\rho_1 - 2eb_2 t_2 \rho_1 - p_1 \rho_1^2 + (1 - r_2)t_1 \rho_1^2,$$
$$N_2 = cp_1 \rho_1^2 + ap_1 \rho_1^2 - b_2 p_1^2 \rho_1^2 + (c + a)r_2 t_1 \rho_1^2 - b_2 p_1 (r_2 t_1 - t_2)\rho_1^2 - (c + a)t_2 \rho_1^2 \tag{3}$$

There exists a unique optimal solution for planning cycle MTO limit $(q_1^*)$ and safety stock level $(q_2^*)$, which are determined by solving the simultaneous Equations (2) and (3).

Analyzing the solution of the simultaneous Equations (2) and (3), we observe that the planning cycle length $(q_1^*/\rho_1)$ is positively related to the per-unit time holding cost and per-unit shortage cost and negatively related to the MTO production rate and price corresponding to zero lead-time. A low safety stock level and MTO production rate (and hence, ability to meet demand), along with a low price corresponding to zero lead-time (and hence, high demand volume), shall make the firm less lead-time competitive.

Product Demand Is Dependent Only on the Guaranteed Lead-Time

In Section 2.2, the problem is modeled considering both lead-time and price-sensitive customers. In this section, we present a model for the case when customers are only lead-time sensitive $(D' = a - 2b_1 T + D)$ by considering price sensitivity negligible $(b_2 = 0)$. Moreover, by setting $e = 0$, we consider $p = p_1$. This model captures the trade-off between lead-time and safety stock. The objective is to maximize the expected profit per-unit time.

Substituting $b_2 = 0$ and $e = 0$, in Equation (1), the expected profit per-unit time is given as:

$$E[z] = -\frac{1}{2cq_1\rho_1}\big(4b_1{}^2 q_1^2 r_2 t_1 + q_1(4b_1(p_1 q_2 - (a + c - q_2)r_2 t_1 + q_1(p_1 + (-1 + r_2)t_1))$$
$$+q_2(-4b_1 + q_2 r_3)t_2)\rho_1 + ((a + c)^2 r_2 t_1 + q_1^2(p_1 + (-1 + r_2)t_1)$$
$$-2q_1(a + c - q_2)(p_1 + (-1 + r_2)t_1) + q_2^2(p_1 + r_2 t_1) - 2(a + c)q_2(p_1 + r_2 t_1 - t_2))\rho_1^2\big) \tag{4}$$

**Lemma 2.** *The expected profit function E[z] is concave in $q_1$ and $q_2$ under certain conditions. The proof of the Lemma 2 is given in the Appendix A.*

The following Proposition addresses the optimal solution of the problem through the necessary conditions of Optimization Theory.

**Proposition 2.** *In the lead-time sensitive market, the manufacturer's optimal MTO production limit and safety stock level $(q_1^*, q_2^*)$ satisfy first-order necessary conditions $\frac{\partial z}{\partial q_1} = 0$ and $\frac{\partial z}{\partial q_2} = 0$ [31].*
*$\frac{\partial z}{\partial q_1} = 0$ and $\frac{\partial z}{\partial q_2} = 0$ represents Equations (3) and (4) respectively as given below:*

$$q_1^2\big(\rho_1(p_1 - t_1 + r_2 t_1)(4b_1 + \rho_1) + 4b_1^2 r_2 t_1\big) - q_2^2 \rho_1^2(p_1 + r_2 t_1)$$
$$+2q_2 \rho_1^2(a + c)(p_1 - t_2 + r_2 t_1) - (a + c)^2 \rho_1^2 r_2 t_1 = 0 \tag{5}$$

$$q_1\big((p_1 + r_2 t_1)(2b_1 + \rho_1) - 2b_1 t_2 - \rho_1 t_1\big) + q_2 \rho_1(p_1 + r_2 t_1) + q_1 q_2 r_3 t_2 + (a + c)t_2 - \rho_1(a + c)(p_1 + r_2 t_1) = 0 \tag{6}$$

There exists a unique optimal solution for MTO production limit $(q_1^*)$ and safety stock level $(q_2^*)$, which are determined by solving the simultaneous Equations (5) and (6).

Analyzing the solution of the simultaneous Equations (5) and (6), we observe that the planning cycle length $(q_1^*/\rho_1)$ is positively related to the per-unit holding cost and per-unit shortage cost, and negatively related to production rate. A low safety stock level and MTO production rate (and hence, ability to meet demand) shall make the firm less lead-time competitive.

## 3. Results

The following data set is used to validate the model. $a = 300$, $D \sim U(0, 400)$, $b_1 = 2$, $b_2 = 0.6$, $e = 0.2$, $p_1 = 130$, $r_2 = 0.4$, $r_3 = 0.04$, $t_1 = 105$, $t_2 = 96$, and $\rho_1 = 10$. The existing literature on lead-time guarantee, namely, Kuthambalayan et al. [21] and Kuthambalayan and Bera [23], has motivated the assignment. Using these parameter values, we obtain the expected profit. Simultaneous equations are obtained through necessary conditions [31] and solved for critical points. After that, sufficient conditions for optimality are satisfied at the critical point, with Hessians (see Appendix A) $H_1 = -0.0085$, $H_2 = -0.0228$, and $H_3 = 0.00012$ for the lead-time and price sensitive customer segment. Similarly, the sufficient conditions for optimality are satisfied at the critical point, with $H_1 = -0.0096$, $H_2 = -0.0226$, and $H_3 = 0.00013$ for the lead-time sensitive customer segment. The expected profit function is concave, and at these parameter values, the optimal values of decision variables (critical point) are $q_1{}^* = 302.58$, $q_2{}^* = 57.46$, $p^* = 117.9$, and $T^* = 30.25$, with $E[z] = 79.36$ for the lead-time and price sensitive customer segment. In a similar way, the expected profit function is concave, and at these parameter values, the optimal values of decision variables (critical point) are $q_1{}^* = 331.38$, $q_2{}^* = 69.92$, and $T^* = 66.26$, with $E[z] = 217.995$ for the lead-time sensitive customer segment. The software Wolfram Mathematica was used throughout. We find the closed-formed solution using the Calculus of Multivariable Optimization theorem [31].

The sensitivity analysis of some key parameters, namely, (1) lead-time sensitivity of the demand (see Figures 1 and 2), (2) price sensitivity of the demand (see Figure 3), (3) lead-time sensitivity of price (see Figure 4), (4) per-unit cost of MTO product (see Figures 5 and 6), (5) per-unit cost of safety stock (see Figures 7 and 8), (6) per-unit per-unit time cost of holding (see Figures 9 and 10), and (7) production rate (see Figures 11 and 12) are conducted. We use another software package, General Algebraic Modeling System (GAMS), for sensitivity analysis. The effect on profit and the decision variables, namely, planning cycle MTO limit, safety stock level, and price, is recorded.

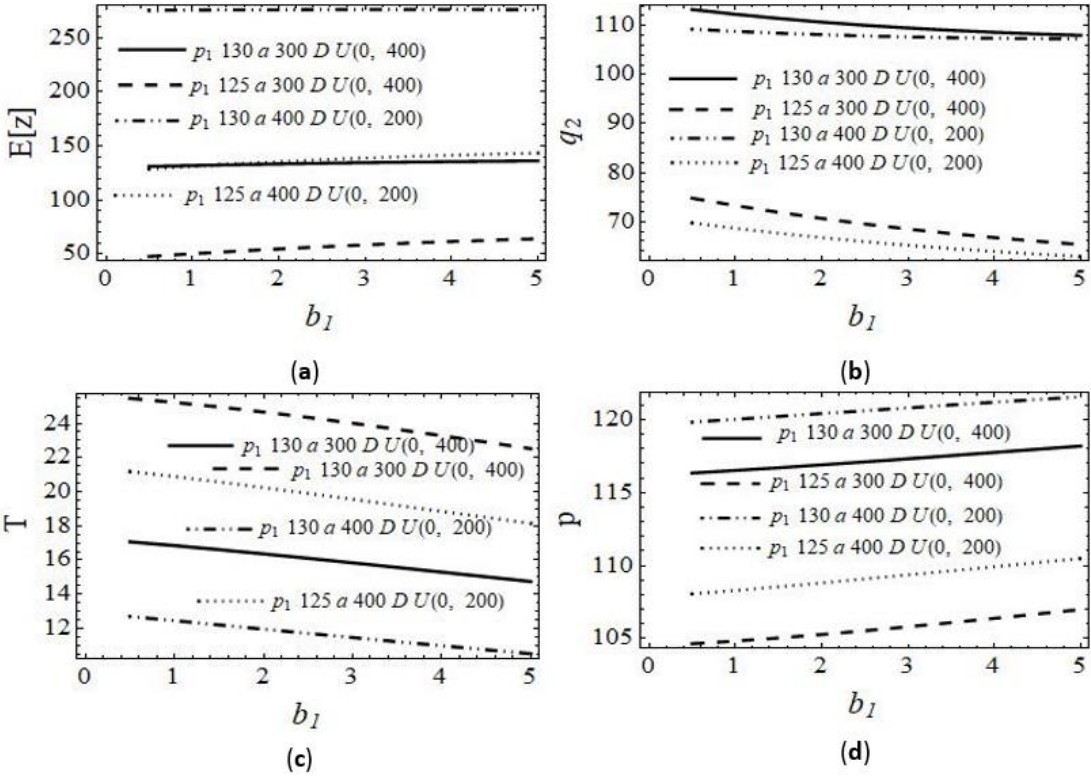

**Figure 1.** Change in (**a**) profit, (**b**) $q_2{}^*$, (**c**) $T^*$, and (**d**) $p^*$ with change in $b_1$ for demand as $f(T, p)$ at $[b_2 = 0.6, e = 0.4, t_1 = 105, t_2 = 95, r_2 = 0.2, r_3 = 0.04, \rho_1 = 10]$.

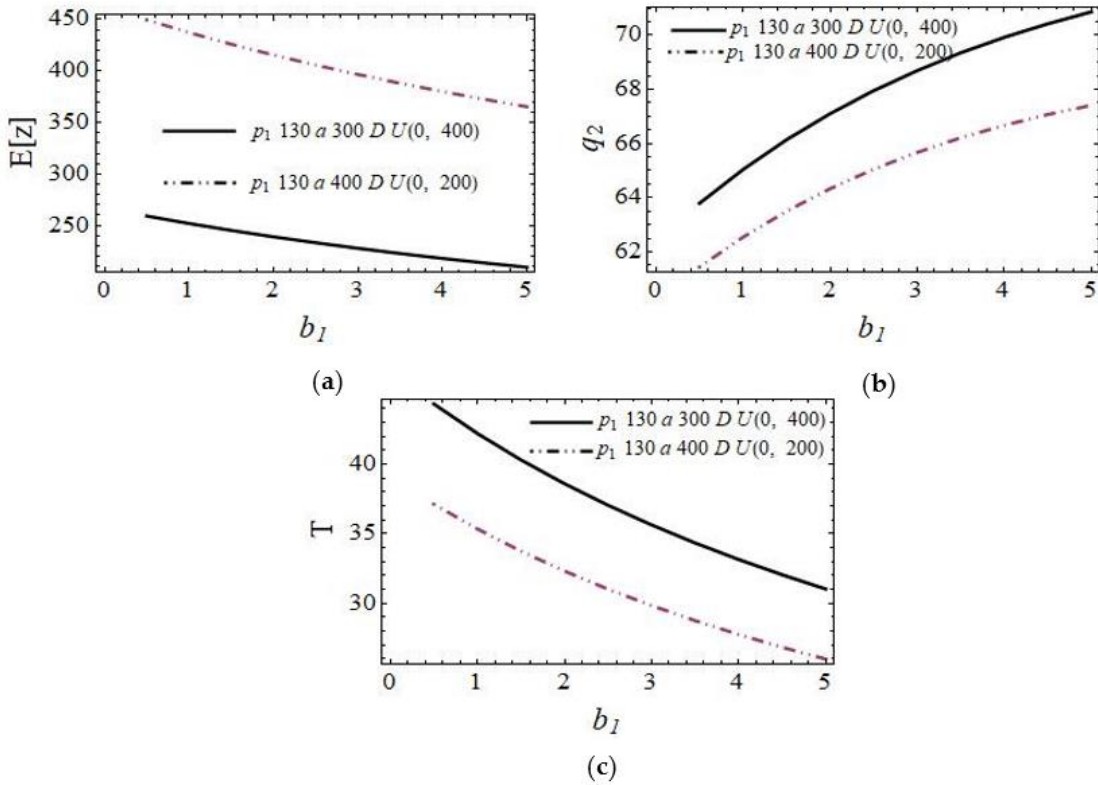

**Figure 2.** Change in (**a**) profit, (**b**) $q_2{}^*$, and (**c**) $T^*$ with change in $b_1$ for demand as *f(T)* at $[t_1 = 105, t_2 = 96, r_2 = 0.4, r_3 = 0.04, \rho_1 = 10]$.

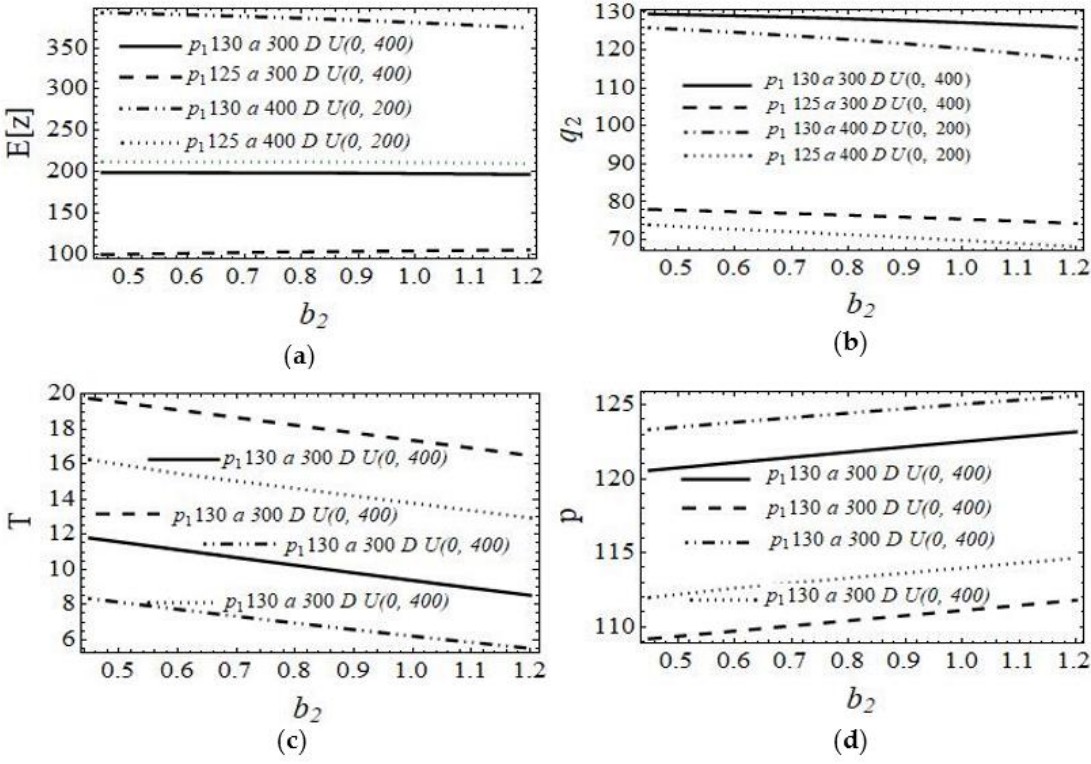

**Figure 3.** Change in (**a**) profit, (**b**) $q_2{}^*$, (**c**) $T^*$, and (**d**) $p^*$ with change in $b_2$ for demand as *f(T, p)* at $[b_1 = 4, e = 0.4, t_1 = 105, t_2 = 95, r_2 = 0.2, r_3 = 0.04, \rho_1 = 12]$.

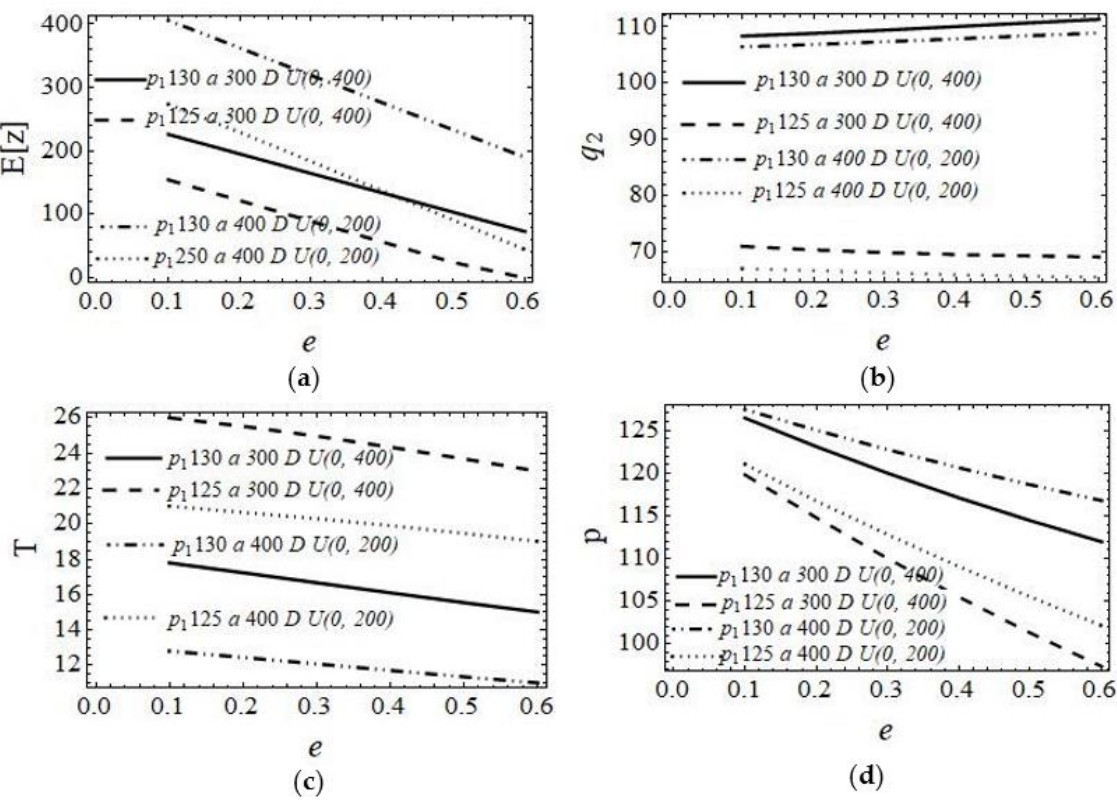

**Figure 4.** Change in (**a**) profit, (**b**) $q_2{}^*$,(**c**) $T^*$, and (**d**) $p^*$ with change in $e$ for demand as $f(T, p)$ at $[b_1 = 2.5, b_2 = 0.6, t_1 = 105, t_2 = 95, r_2 = 0.2, r_3 = 0.04, \rho_1 = 10]$.

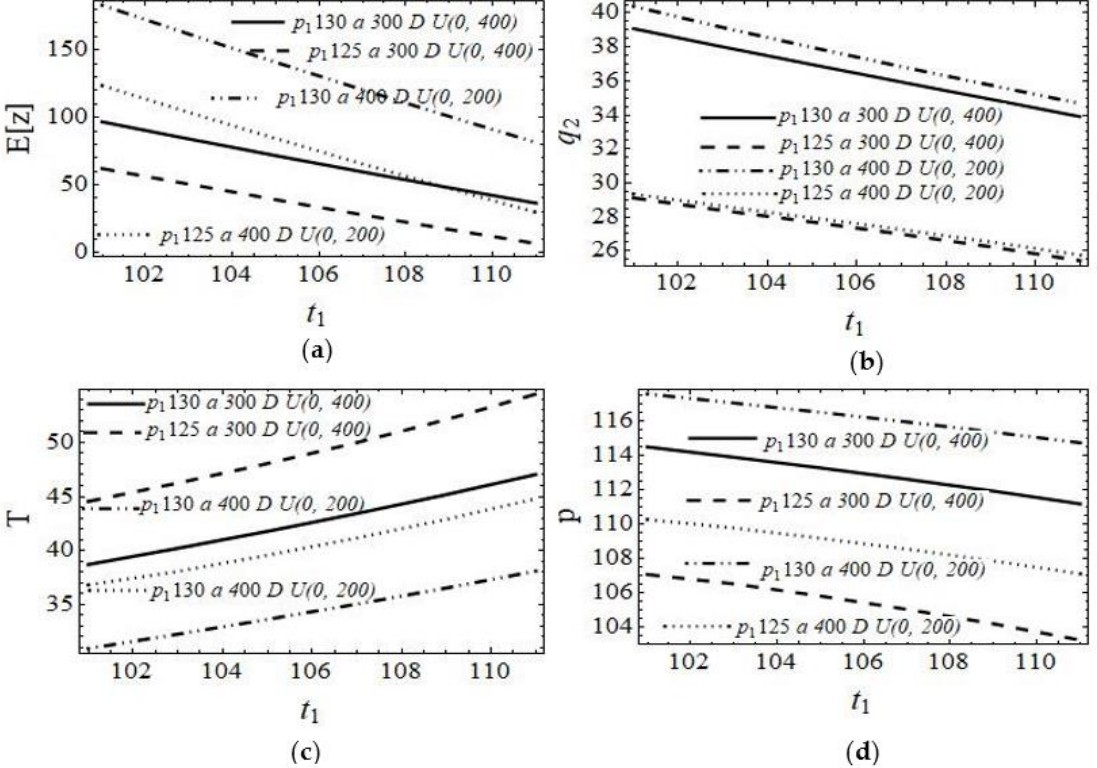

**Figure 5.** Change in (**a**) profit, (**b**) $q_2{}^*$, (**c**) $T^*$, and (**d**) $p^*$ with change in $t_1$ for demand as $f(T, p)$ at $[b_1 = 2, b_2 = 0.6, e = 0.2, t_2 = 90, r_2 = 0.2, r_3 = 0.1, \rho_1 = 6]$.

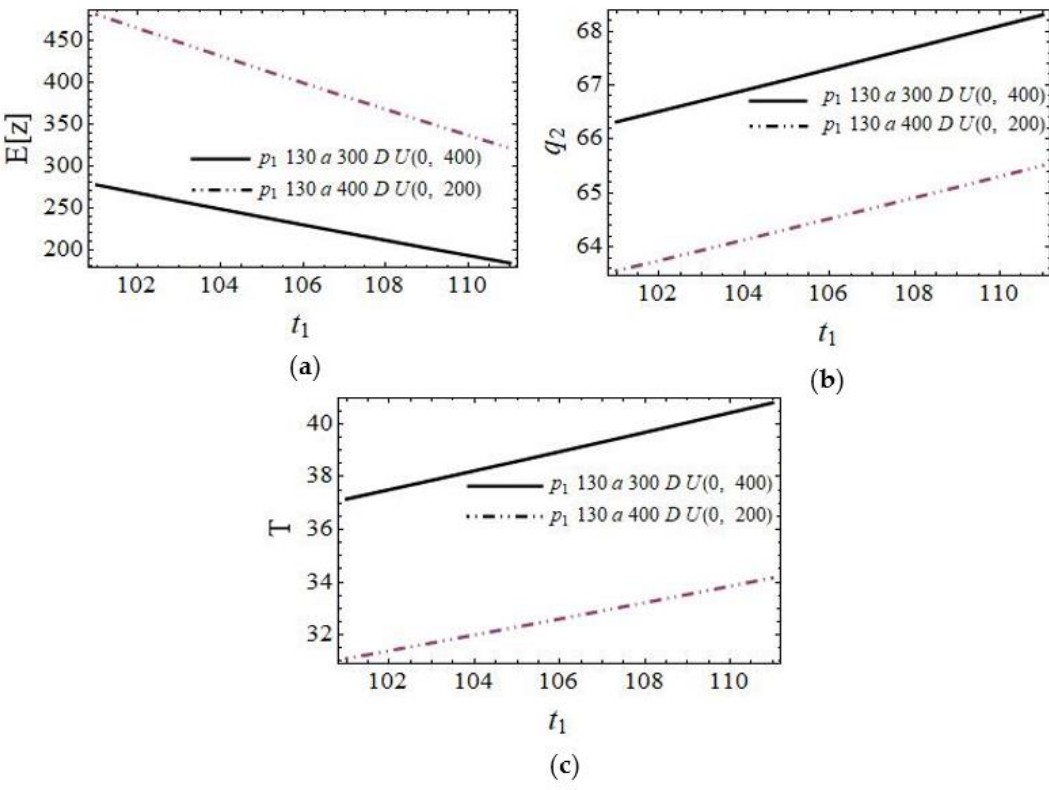

**Figure 6.** Change in (**a**) profit, (**b**) $q_2^*$, and (**c**) $T^*$ with change in $t_1$ for demand as $f(T)$ at $[b_1 = 2, t_2 = 96, r_2 = 0.4, r_3 = 0.04, \rho_1 = 10]$.

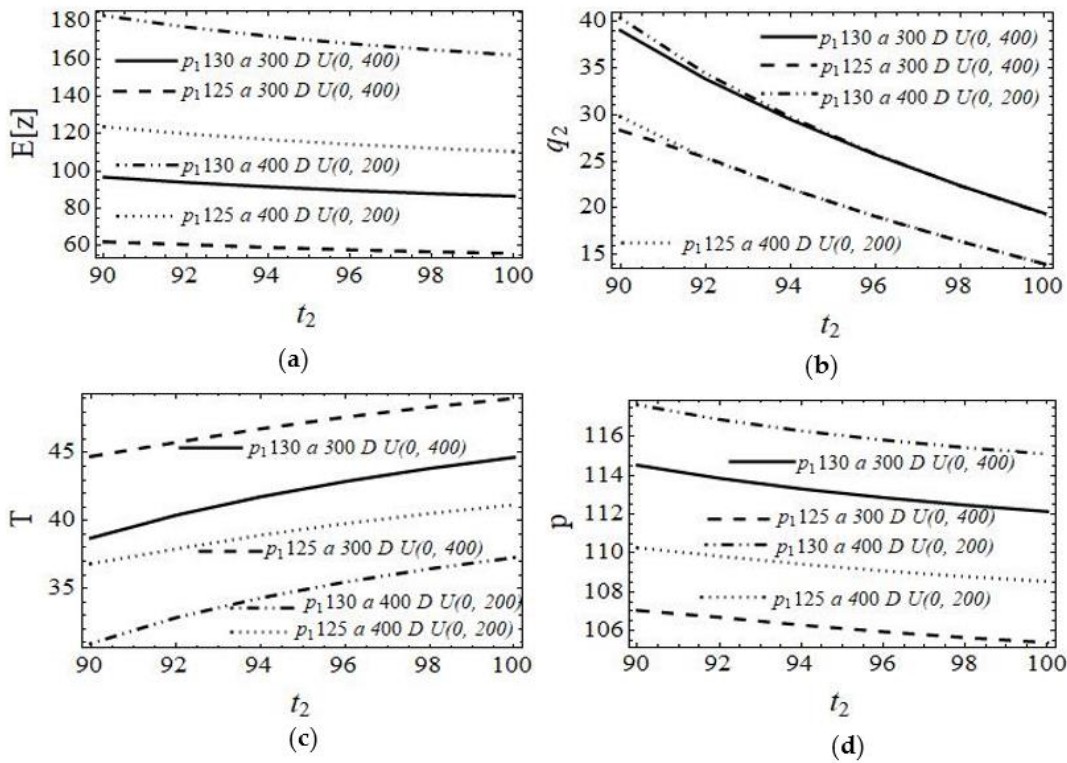

**Figure 7.** Change in (**a**) profit, (**b**) $q_2^*$, (**c**) $T^*$, and (**d**) $p^*$ with change in $t_2$ for demand as $f(T, p)$ at $[b_1 = 2, b_2 = 0.6, e = 0.2, t_1 = 101, r_2 = 0.2, r_3 = 0.1, \rho_1 = 6]$.

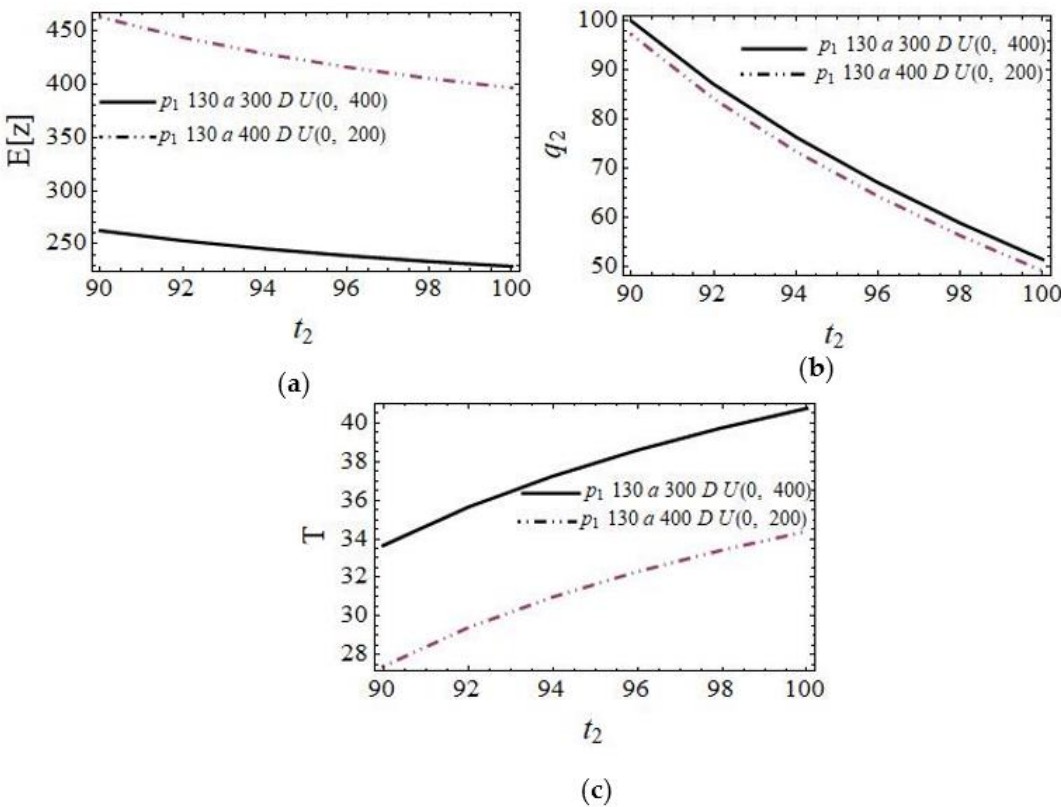

**Figure 8.** Change in (**a**) profit, (**b**) $q_2{}^*$, and (**c**) $T^*$ with change in $t_2$ for demand as *f(T)* at $[b_1 = 2, t_1 = 105, r_2 = 0.4, r_3 = 0.04, \rho_1 = 10]$.

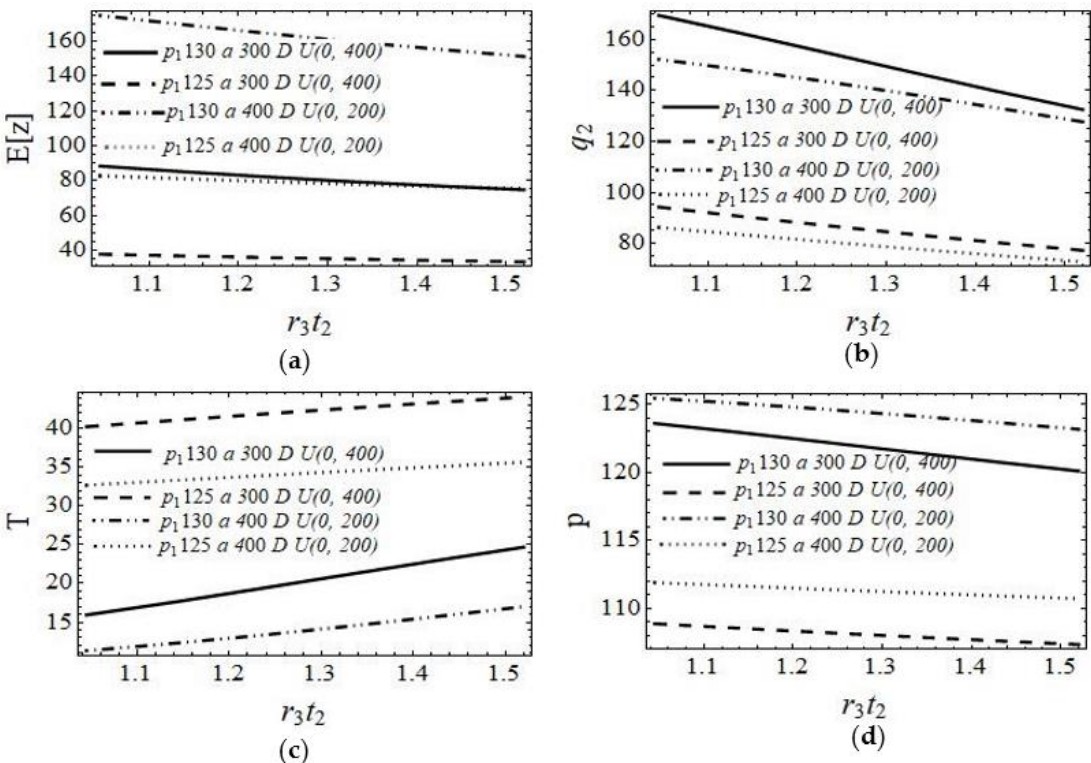

**Figure 9.** Change in (**a**) profit, (**b**) $q_2{}^*$, (**c**) $T^*$, and (**d**) $p^*$ with change in holding cost for demand as *f(T, p)* at $[b_1 = 2, b_2 = 0.6, e = 0.2, t_1 = 105, t_2 = 95, r_2 = 0.2, \rho_1 = 5]$.

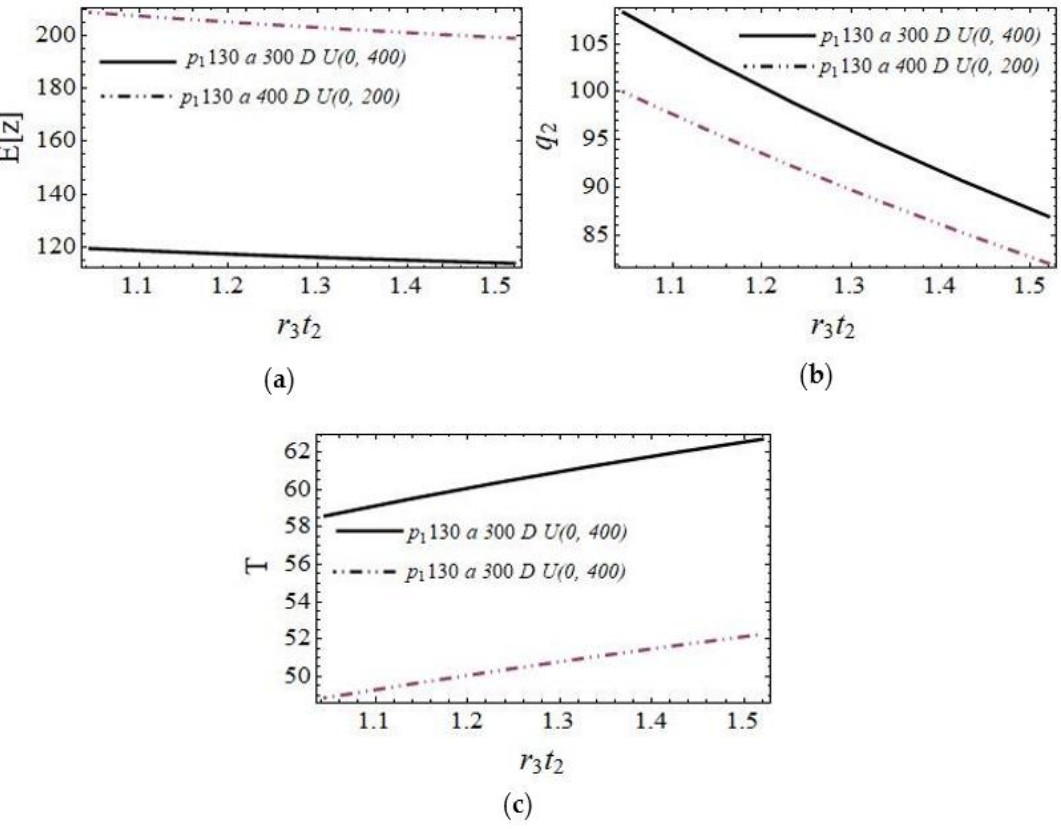

**Figure 10.** Change in (**a**) profit, (**b**) $q_2{}^*$, and (**c**) $T^*$ with change in holding cost for demand as *f(T)* at $[b_1 = 2, t_1 = 105,\ t_2 = 95, r_2 = 0.4, \rho_1 = 5]$.

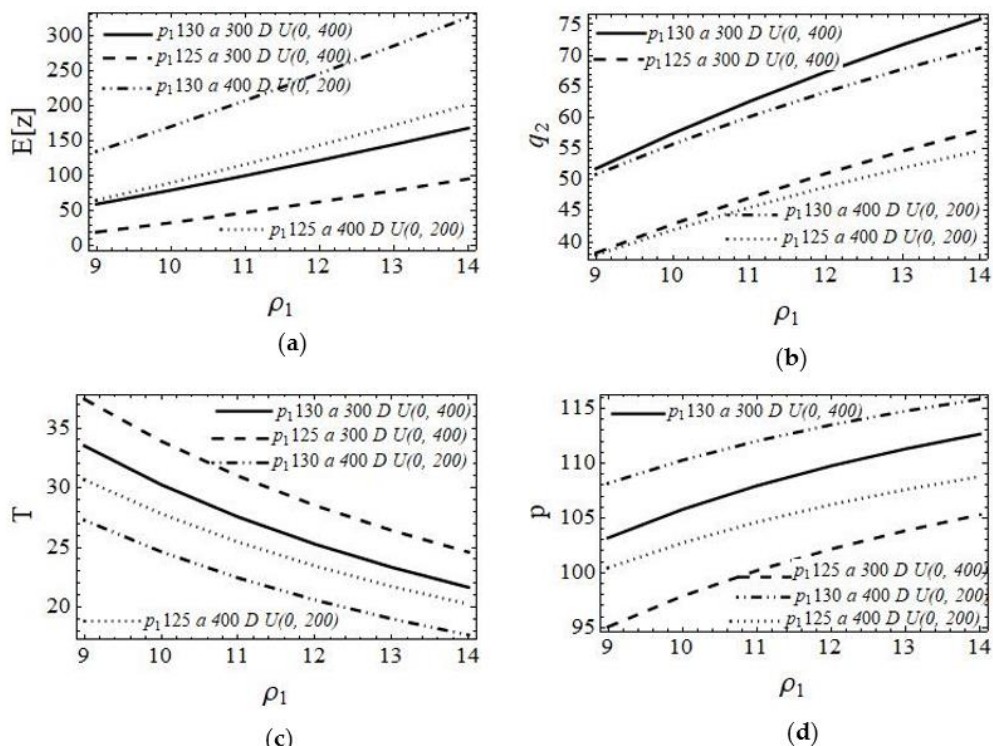

**Figure 11.** Change in (**a**) profit, (**b**) $q_2{}^*$, (**c**) $T^*$, and (**d**) $p^*$ with change in $\rho_1$ for demand as *f(T, p)* at $[b_1 = 4, b_2 = 0.6, e = 0.4, t_1 = 105, t_2 = 96, r_2 = 0.4, r_3 = 0.04, \rho_1 = 10]$.

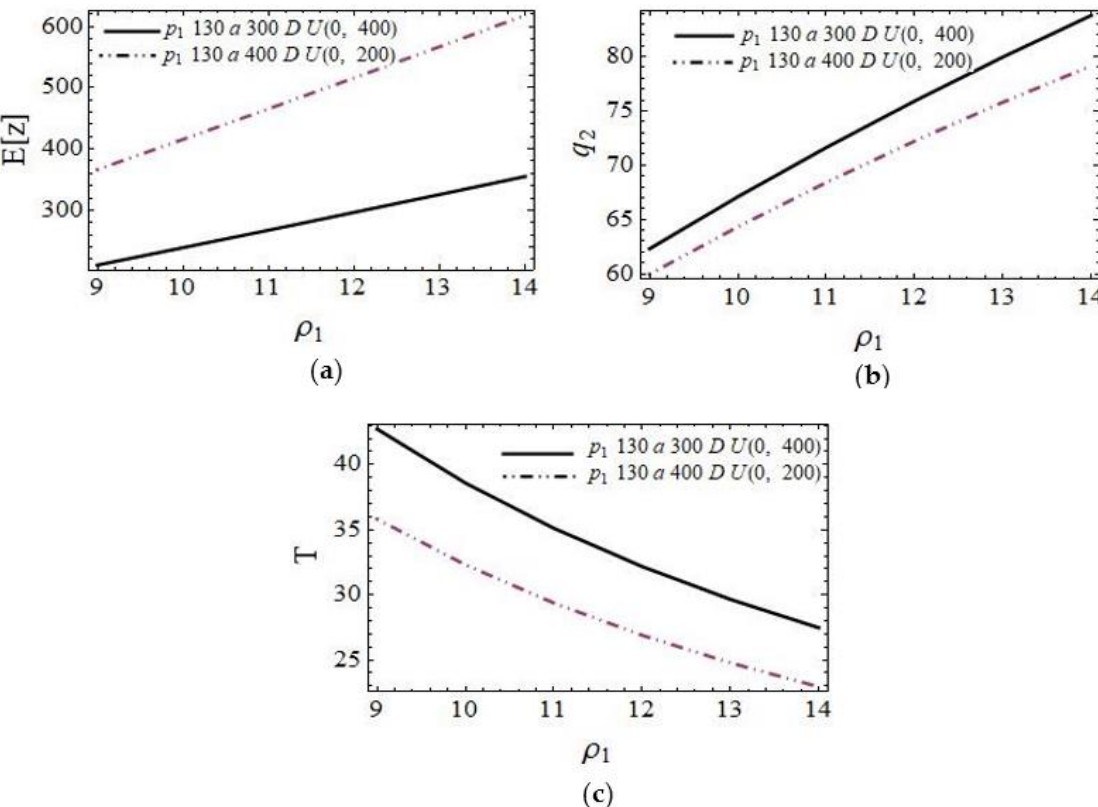

**Figure 12.** Change in (**a**) profit, (**b**) $q_2{}^*$, and (**c**) $T^*$ with change in $\rho_1$ for demand as *f(T)* at $[b_1 = 2, t_1 = 105, t_2 = 96, r_2 = 0.4, r_3 = 0.04, \rho_1 = 10]$.

## 4. Discussion

From the sensitivity analysis, the following observations are obtained.

### 4.1. Effect of Holding Cost

When the per-unit time holding cost is fixed at a meagre value, the firm meets demand using safety stock and guarantees a low lead-time in both the cases $D' = f(T, p)$ and $D' = f(T)$ (Figures 9 and 10). Both figures show that with an increase in the holding cost, the safety stock level decreases (Figures 9b and 10b), and the MTO production limit and lead-time increase (Figures 9c and 10c). The increase in lead-time negatively affects the demand volume and is counteracted by decreasing the selling price of the product (Figure 6d). Consequently, the firm's profit decreases with increasing holding costs.

Managers should treat holding costs as a policy variable [32]. Its value should ensure that the corresponding lead-time guarantee is competitive (i.e., compares well with the lead-time promised by the firm's competitor) and meets the customers' expectations. In an integrated decision model, this lead-time quote is based on the firm's operational constraints (e.g., MTO production rate).

### 4.2. Effect of Production Costs on Decision Variables

Reducing a product's cost increases the profit margin and the firm's profit. It is the outcome of an improvement in the firm's Total Productivity without compromising on quality and service and is a vital concern of an organization. A higher lead-time guarantee due to sourcing less costly materials from low-cost countries can sometimes be more profitable [33]. The holding cost value is usually directly proportional to the product's cost ($t_1$, $t_2$), and a corresponding reduction in holding cost causes the safety stock level to increase (Figures 5, 6, 7b and 8b), thereby reducing the lead-time and MTO production limit. A decrease in MTO production limit implies a decrease in the lead-time (Figures 5, 6, 7c and 8c).

A lead-time decrease has a positive effect on demand volume, causing the firm's profit to increase (Figures 5, 6, 7a and 8a), thereby allowing the firm to quote a higher price when the price is lead-time sensitive.

### 4.3. Effect of Market Characteristics on Decision Variables

#### 4.3.1. Effect of $b_1$

When the product demand is dependent on lead-time and price ($D' = f(T, p)$), as the lead-time sensitivity of demand ($b_1$) increases, to compete effectively, the lead-time decreases (Figure 1c). The decrease in demand volume due to the increase in the lead-time sensitivity of demand ($b_1$) is thus counteracted to an extent by the lead-time decrease. Similarly, in the only lead-time market ($D' = f(T)$), with an increase in lead-time sensitivity of demand ($b_1$), the lead-time decreases (Figure 8c). The MTO production limit decreases with lead-time. However, the safety stock increases (Figures 1b and 2b) and the profit decreases (Figures 1a and 2a) only when the product demand is dependent solely on lead-time. Jayaram et al. [34] determined that financial performance is strongly related to competitive lead-time, based on a study of three car manufacturers in North America. Tiedemann et al. [35] confirmed this based on multiple case studies.

When the price is dependent on the lead-time, with reduced lead-time, at the higher value of lead-time sensitivity of the price ($e = 0.4$), the firm, instead of increasing the safety stock level (Figure 3b), increases the selling price (Figure 1d) and profit increases (Figure 1a). The increase in selling price negatively affects the demand volume, thereby controlling the lead-time decrease and the safety stock increase. Therefore, the study suggests that the firms that are lead-time competitive can sell the product in markets where the lead-time sensitivity of the price is high, i.e., the customers are willing to pay more for reduced lead-time.

#### 4.3.2. Effect of $b_2$

Given that $a' > 0$ and $b' > 0$, the situation under study is when customers are more lead-time sensitive than price sensitive. $b' \leq 0$, however, shall imply that the customers are willing to wait to obtain price benefits, and the *demand volume* increases or remains unchanged with *the lead-time* [14]. With an increase in price sensitivity of demand ($b_2$), the demand volume decreases, which is counteracted by a lead-time decrease (Figure 3c), reducing the MTO production limit. Despite the reduced lead-time, the safety stock decreases (Figure 3b) at a higher value of $e$ (=0.4) and is explained by the price increase (Figure 3c), reducing the demand volume. The profit obtained also decreases (Figure 3a) because of decrease in demand at a high value of $e$ (=0.4).

#### 4.3.3. Effect of $e$

With an increase in lead-time sensitivity of price €, the price quoted decreases and is counteracted by a lead-time reduction, reducing the planning cycle MTO limit and increasing the safety stock level. The profit obtained also decreases because of the decrease in price (Figure 4a). Thus, it is evident that a lead-time reduction offers the possibility of a price premium besides increasing demand. In a time-based competition, e-retailers are known to set prices based on lead-time (quicker delivery at a higher charge), and the lead-time is uniform to all potential customers [6].

### 4.4. Effect of Production Rate

In the past few decades, many production systems have focused on reducing inventories by adopting just-in-time sourcing (where the firm's suppliers provide goods based on end-customer orders) [36]. The firm has spare production capacity or can vary the production rate to compensate for the low inventory stocks and be lead-time competitive in the face of variations in system load. The drawback is the increased risk of production and delivery failure the firm faces when its supply (raw material/semi-finished goods) is disrupted. The firm's operational constraints are loosened with an increase in the MTO

production rate, reducing planning cycle length (Figures 11c and 12c) and lead-time. The safety stock level (Figures 11b and 12b), product price (Figure 11d), and expected profit (Figures 11a and 12a) increase.

### 4.5. Effect of Demand Variation

A change from $a = 400$, $D = U(0, 200)$ to $a = 300$, $D = U(0, 400)$ reflects an increase in demand variance. Demand variability negatively impacts profit per-unit time due to the increase in aggregate costs (Figure 1a, Figure 2a, Figure 3a, Figure 4a, Figure 5a, Figure 6a, Figure 7a, Figure 8a, Figure 9a, Figure 10a, Figure 11a, Figure 12a). To counteract this, the length of the planning cycle length (hence, the lead-time) is increased (Figure 1c, Figure 2c, Figure 3c, Figure 4c, Figure 5c, Figure 6c, Figure 7c, Figure 8c, Figure 9c, Figure 10c, Figure 11c, Figure 12c). This step lowers the demand volume and provides additional time to MTO. The lead-time increase also results in a reduced price quote (Figure 1d, Figure 2d, Figure 3d, Figure 4d, Figure 5d, Figure 6d, Figure 7d, Figure 8d, Figure 9d, Figure 10d, Figure 11d). The increase in demand variability leads to an increase in safety stock level when the holding cost is set low (Figure 1b, Figure 2b, Figure 3b, Figure 4b, Figure 8b, Figure 9b, Figure 10b, Figure 11b, Figure 12b), but may lead to its reduction, increasing the preference towards made-to-order, when set high (Figures 5 and 7b). It is not, therefore, always advisable to increase the safety stock level in response to an increase in demand variability, as in a pure make-to-stock system. This counterintuitive observation results from the complex trade-offs in the integrated operations-marketing model.

### 4.6. Effect of $p_1$

Consumers do not postpone their purchase decision during festive seasons, e.g., Christmas. They are sensitive to lead-time and willing to pay a price premium for the delivery on their chosen date, prompting firms to hold higher stocks. A change from $p_1 = 125$ to $p_1 = 130$ implies a higher selling price for a given lead-time and reflects a higher opportunity to charge a price premium for a given lead-time decrease. The setting $p_1 = 130$ thereby resulted in a higher price (Figure 1d, Figure 2d, Figure 3d, Figure 4d, Figure 5d, Figure 6d, Figure 7d, Figure 8d, Figure 9d, Figure 10d, Figure 11d) and increased profit per-unit time (Figure 1a, Figure 2a, Figure 3a, Figure 4a, Figure 5a, Figure 6a, Figure 7a, Figure 8a, Figure 9a, Figure 10a, Figure 11a). The resulting reduction in demand volume is countered by quoting a shorter lead-time guarantee (Figure 1c, Figure 2c, Figure 3c, Figure 4c, Figure 5c, Figure 6c, Figure 7c, Figure 8c, Figure 9c, Figure 10c, Figure 11c) and increasing the safety stock level (Figure 1b, Figure 2b, Figure 3b, Figure 4b, Figure 5b, Figure 6b, Figure 7b, Figure 8b, Figure 9b, Figure 10b, Figure 11b) to improve lead-time competitiveness.

### 5. Conclusions

In this study, the supply–demand mismatch is reduced through (1) product management (postponement, e.g., MTO), (2) demand management (using three product attributes- lead-time guarantee, price, and quality, as levers), (3) information management (to improve inter-department coordination), and (4) supply management (using secondary supply source to stock safety stocks). A reliable lead-time promise reduces the customer's supply risk and is advertised by the firm to capture demand. Past studies on the lead-time guarantee in an MTO production system ignore (1) safety stocks and (2) the lead-time-sensitive price.

The problem is modeled as an unconstrained stochastic non-linear programming problem of two stages, maximizing the expected profit per-unit of time. In Stage 1, before demand realization, operations decide the level of safety stocks and length of the production planning cycle, thereby managing supply. Marketing decides the three product attributes, thereby managing demand. The price is explicitly modeled as a function of the lead-time guarantee. In Stage 2, demand is realized and is modeled as a linear function of the product attributes, with a non-negative stochastic error component. The integrated operations-

marketing model captures the trade-offs between lead-time guarantee, price, and safety stock. The multivariable optimization technique provides a closed-form solution.

The sensitivity analysis of key problem parameters reveals that lead-time competitiveness is adversely affected by a low safety stock level, MTO production rate (i.e., low supply capability), and product price (i.e., high demand volume). To promise a short lead-time, an integrated firm aims to increase safety stocks by reducing product costs and setting a low inventory holding cost. A higher price quote corresponding to a shorter lead-time in a lead-time sensitive market negatively affects demand volume and reduces the need to increase the safety stock level. In a price-sensitive market, the firm would reduce the lead-time rather than the price. Demand variability is countered by guaranteeing a longer lead-time and increasing the safety stock (The stock level is lowered when the holding costs are high). The firm sets a higher product price and quotes a shorter lead-time (thereby increasing the safety stock level) in response to an opportunity to charge a higher price premium for a given lead-time decrease.

The hierarchical operation-marketing model, which sequentially determines the marketing decisions (price and lead-time guarantee) and operations decisions (safety stock), shall provide a suboptimal solution [37]. In a hierarchical decision-making approach, the outcome of the marketing decisions constrains the operations decisions. The problem suboptimality resulting from a hierarchical approach increases with the problem size [38]. The integrated operation-marketing model promotes joint decision-making [39] and produces better solutions [40].

This study highlights the importance of integrating operations decisions on inventory and production cycle time, marketing decisions on price, lead-time, and understanding the trade-offs that lead to improved decisions. Future research could consider the stochastic production rate and an assembly system with component commonality.

**Author Contributions:** Conceptualization, T.S.K.; methodology, T.S.K.; software, S.K.D.; validation, S.K.D.; formal analysis, S.K.D.; investigation, S.K.D.; data curation, S.K.D.; writing—original draft preparation, T.S.K.; writing—review and editing, T.S.K.; visualization, S.K.D.; supervision, T.S.K. All authors have read and agreed to the published version of the manuscript.

**Funding:** No funding was received for conducting this study.

**Data Availability Statement:** Not applicable.

**Acknowledgments:** Both the authors thank the anonymous referees and the editor for their valuable feedback, which significantly improved the positioning and presentation of this paper.

**Conflicts of Interest:** The authors declare no conflict of interest.

## Appendix A

Proof of Lemma 1: From Equation (1), we get the following:

$$\frac{\partial^2 z}{\partial q_1^2} = \frac{1}{cq_1^3 \rho_1}(8e(b_1 - eb_2)q_1^3 + 2eq_1^3\rho_1 - ((c + a - q_2)^2 r_2 t_1 + b_2 p_1^2(2q_2 + b_2 r_2 t_1)$$
$$+ 2(c + a)q_2 t_2 + p_1(q_2^2 - 2(c + a)b_2 r_2 t_1 - 2q_2(c + a + b_2(-r_2 t_1 + t_2))))\rho_1^2) \le 0$$

$$or \; 8e(b_1 - eb_2)q_1^3 + 2eq_1^3\rho_1 - ((c + a - q_2)^2 r_2 t_1 + b_2 p_1^2(2q_2 + b_2 r_2 t_1)$$
$$+ 2(c + a)q_2 t_2 + p_1 \begin{pmatrix} q_2^2 - 2(c + a)b_2 r_2 t_1 \\ -2q_2(c + a + b_2(-r_2 t_1 + t_2)) \end{pmatrix})\rho_1^2 \le 0 \qquad \text{(A1)}$$

$$\frac{\partial^2 z}{\partial q_2^2} = -\frac{q_1(r_3 t_2 - 2e) + (p_1 + r_2 t_1)\rho_1}{cq_1} \le 0$$

$$H = \begin{vmatrix} \frac{\partial^2 z}{\partial q_1^2} & \frac{\partial^2 z}{\partial q_2 \partial q_1} \\ \frac{\partial^2 z}{\partial q_1 \partial q_2} & \frac{\partial^2 z}{\partial q_2^2} \end{vmatrix} \geq 0; \ or$$

$$\frac{1}{c^2 q_1^4 \rho_1^2}\left(-\left(4e(b_1 - eb_2)q_1^2 + 2eq_1^2\rho_1 + \left(\begin{array}{c}(-c - a + b_2 p_1 + q_2)(p_1 + r_2 t_1) \\ + (c + a - b_2 p_1)t_2\end{array}\right)\rho_1^2\right)^2\right.$$
$$+\rho_1(q_1(2e - r_3 t_2) - (p_1 + r_2 t_1)\rho_1)(8e(b_1 - eb_2)q_1^3 + 2eq_1^3\rho_1 -$$
$$((c + a - q_2)^2 r_2 t_1 + b_2 p_1^2(2q_2 + b_2 r_2 t_1) + 2(c + a)q_2 t_2 +$$
$$\left. p_1\left(q_2^2 - 2(c + a)b_2 r_2 t_1 - 2q_2(c + a + b_2(-r_2 t_1 + t_2)))\right)\rho_1^2)\right) \geq 0$$

Since, under mentioned conditions $\frac{\partial^2 z}{\partial q_1^2} \leq 0\,(H_1 < 0)$, $\frac{\partial^2 z}{\partial q_2^2} \leq 0\,(H_2 < 0)$ and $2 \times 2$ Hessian $(H_3 > 0)$ is negative definite for all $q_1$ and $q_2$. Thus, the profit function is concave under the conditions and the optimal MTO $(q_1^*)$ and safety stock $(q_2^*)$ production quantities are determined by solving the simultaneous equation $\frac{\partial z}{\partial q_1} = 0$ and $\frac{\partial z}{\partial q_2} = 0$

Proof of Lemma 2: From Equation (4), we get the following:

$$\frac{\partial^2 z}{\partial q_1^2} = -\frac{\left((a+c)^2 r_2 t_1 + q_2^2(p_1 + r_2 t_1) - 2(a+c)q_2(p_1 + r_2 t_1 - t_2)\right)\rho_1}{cq_1^3} \leq 0$$
$$or \ (a+c)^2 r_2 t_1 + q_2^2(p_1 + r_2 t_1) \geq 2(a+c)q_2(p_1 + r_2 t_1 - t_2)$$

$$\frac{\partial^2 z}{\partial q_2^2} = -\frac{q_1 r_3 t_2 + (p_1 + r_2 t_1)\rho_1}{cq_1} \leq 0$$

$$H = \begin{vmatrix} \frac{\partial^2 z}{\partial q_1^2} & \frac{\partial^2 z}{\partial q_2 \partial q_1} \\ \frac{\partial^2 z}{\partial q_1 \partial q_2} & \frac{\partial^2 z}{\partial q_2^2} \end{vmatrix} \geq 0 \ or$$
$$q_1 r_3 \left((a+c)^2 r_2 t_1 + q_2^2(p_1 + r_2 t_1) - 2(a+c)q_2(p_1 + r_2 t_1 - t_2)\right)t_2 \geq$$
$$(a+c)^2\left(\begin{array}{c}r_2 t_1(p_1 - 2t_2) \\ + (p_1 - t_2)^2\end{array}\right)\rho_1$$

Since, under mentioned conditions $\frac{\partial^2 z}{\partial q_1^2} \leq 0\,(H_1 < 0)$, $\frac{\partial^2 z}{\partial q_2^2} \leq 0\,(H_2 < 0)$ and $2 \times 2$ Hessian $(H_3 > 0)$ is negative definite. Thus, the profit equation is concave under the conditions and the optimal MTO $(q_1^*)$ and safety stock $(q_2^*)$ production quantities are determined by solving the simultaneous equation $\frac{\partial z}{\partial q_1} = 0$ and $\frac{\partial z}{\partial q_2} = 0$.

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
