# Peer review of "Matching Supply and Demand with Lead-Time Dependent Price and with Safety Stocks in a Make-to-Order Production System"

_systems, doi:10.3390/systems10060256_

Round 1

Reviewer 1 Report

This study focuses on the lead-time guarantee in a Make-to-Order (MTO) production System and proposes a two-stage unconstrained stochastic nonlinear programming with the objective of maximizing unit time and expected profit. The topic of manuscript is interesting and the logic is clear, but there are some issues with this manuscript. If the authors will overcome the problems, it will be the sound and significant work. Therefore, some suggestions and quarries, which will be helpful to the authors, are given to improve the paper. Specific comments:

1. The abstract section needs to highlight more clearly the research question, the purpose of the study, the results of the study, and the managerial implications of its research. This can highlight the original innovation and value of the study.

2. In the introduction, a background to the research question is missing and some realistic examples could be added to illustrate why the study was conducted. In addition, a brief summary of the findings of the study and its management implications is missing.

3. The number of references is low and needs to be enriched.

4. In Section 2, a detailed description of the study scenario is needed, as well as an addition to the decision sequence.

5. In Section 3, the assignment of parameters needs to be explained and used to justify the assignment.

6. In Section 4, on sensitivity analysis to obtain observations, the authors divided into 4 subsections, but some subsections are too small. It is suggested to enrich the content or summarize it into one section.

7. The authors raise management insights in the abstract, but they are not found in the conclusion section and need to be added.

Reviewer 2 Report

1. There are too few types of experiments, so additional experiments are needed, such as sensitivity analysis.

2. In the discussion part, authors should explain some reality examples based on experimental results.

3. Authors should write both managerial and academic insights of this study.

Round 2

Reviewer 1 Report

This paper study the ability to reduce the supply-demand mismatch of a periodic Make-to-Order (MTO) production system using safety stocks with marketing managing demand using lead-time

guarantee and price as levers. The systematic framework that the author carries out are appropriate to this kind of study. Some comments on this paper appear below.

 1.     The references are not in order, please correct them.

2.     Please use the three-line table for the table 1 in the article.

3.     In Section 3, the font formats of the script are different, and the formula is too large. Please adjust.

4.     Please add the title of sub graph.

5.     The references cited in the article are inconsistent with the list, please check carefully
